# Comparisons between Manual Lymph Drainage, Abdominal Massage, and Electrical Stimulation on Functional Constipation Outcomes: A Randomized, Controlled Trial

**DOI:** 10.3390/ijerph17113924

**Published:** 2020-06-01

**Authors:** Jacqueline S. Drouin, Lucinda Pfalzer, Jung Myo Shim, Seong Jung Kim

**Affiliations:** 1School of Health Sciences, Oakland University, 433 Meadow Brook Road, Rochester, MI 48309-4451, USA; drouin@oakland.edu; 2Physical Therapy Department, University of Michigan-Flint, 2157 WSW Bldg., Flint, MI 48502-195, USA; cpfalzer@umich.edu; 3Department of Skin and Health Care, Suseong University, 15 Dalgubeol-daero 528-gil, Suseong-gu, Daegu 13557, Korea; Shimgo1@naver.com; 4Department of Physical Therapy, College of Health and Science, Kangwon National University, 346, Hwangjo-gil, Dogye-eup, Samcheok-si, Gangwon-do 24341, Korea

**Keywords:** heart rate variability, stress and anxiety, bowel movement, constipation, manual lymph drainage, massage, electrical stimulation

## Abstract

Background: Evidence supports abdominal massage (AM) or electrical stimulation (ES) as effective in treating functional constipation (FC). Manual lymph drainage (MLD) may also be beneficial, however, it was not previously investigated or compared to ES and AM. Methods: Sixteen college-aged males and 36 females were recruited. Participants were randomly assigned to MLD, AM or ES. Heart rate variability (HRV) measures for total power (TP), high frequency (HF), low frequency and LF/HF ratio assessed ANS outcomes. state-trait anxiety inventory (STAI) and stress response inventory (SRI) assessed psychological factors and bowel movement frequency (BMF) and duration (BMD) were recorded daily. Results: MLD significantly improved all ANS measures (p≤0.01); AM significantly improved LF, HF and LF/HF ratios (p = 0.04); and ES significantly improved LF (p = 0.1). STAI measures improved, but not significantly in all groups. SRI improved significantly from MLD (p < 0.01), AM (p = 0.04) and ES (p < 0.01), but changes were not significant between groups. BMD improved significantly in all groups (p≤ 0.02). BMF improved significantly only following MLD and AM (p < 0.1), but differences between groups were not significant (p = 0.39). Conclusions: MLD significantly reduced FC symptoms and MLD had greater improvements than AM or ES.

## 1. Introduction

Evidence supports the utility of abdominal massage (AM) and electrical stimulation (ES) in the treatment of functional constipation (FC) [1,2,3,4]. While manual lymph drainage (MLD) may be beneficial by promoting normal peristalsis and autonomic nervous system balance, ANS balance, these effects have not been previously investigated.

Functional constipation is described as difficult, incomplete and prolonged passage of dry, hardened stools through the bowel [5,6]. Symptoms include pain, abdominal distension, rectal fullness and strain during defecation [6,7,8]. The incidence ranges from 2–28% and varies by region, age and gender and pathologic, behavioral and psychological factors [9]. It is 2.2 times more prevalent among women than men, and occurs more frequently among elderly than younger people [10]. When FC becomes chronic, individuals risk developing physical, mental and social disorders. Physical disorders include hemorrhoids, hernias, anal fissures, colonic muscle fiber laxity, colonic wall thickening and prolapse and headaches [11]. Referred pain may occur when a bolus of hard feces compresses the left iliopsoas muscle and activates myofascial trigger points [11]. Mental and social changes include fatigue, loss of appetite and reductions in social, leisure and work activities diminish the person’s quality of life [12,13].

Constipation may be organic or functional [14]. Organic constipation occurs from pathologic conditions like neoplasms, partial intestinal obstructions, spinal cord compression, metabolic and neurological diseases and pharmacological interventions. Organic constipation often requires medical or pharmacologic management [14,15]. Functional constipation, termed idiopathic, is associated with environmental or behavioral factors like inadequate water or fiber consumption, reduced physical activity or exercise pattern changes and mental stress [14]. Neuromuscular dysfunction of the defecation unit can lead to disordered or difficult defecation as a cause of FC [16]. General treatments for FC include non-pharmaceuticals, pharmaceuticals and surgery. Non-pharmaceutical therapy includes lifestyle adjustments that promote good health habits to facilitate defecation such as dietary changes like increasing fiber and water consumption [17,18,19]. Other non-pharmacologic approaches include reflex-therapy, biofeedback, massage, translumbar and transsacral magnetic stimulation–induced motor evoked potentials and electrical stimulation [20,21,22]. When FC becomes chronic, like OC, it may require pharmacologic interventions [23]. Pharmaceuticals mitigate symptoms, however once discontinued, FC tends to recur [24]. Further, prolonged pharmaceutical usage may aggravate constipation and cause side-effects like fecal impaction and bowel perforation [25]. Long-term pharmaceutical use may prevent the biome from cleaning toxins from the colon which increase colorectal cancer risk [26]. Surgery is used in rare cases and is not a routine treatment [27]. Thus, non-pharmaceutical treatments are safe and most preferred, with AM and ES the most prevalent.

Abdominal massage was used for several centuries to treat constipation [1,28]. It was popular in the early 20th century and a key component in PT curricula in many countries until its use declined by the 1950s. Interest in AM rose again in oncology and hospice and palliative care environments, due to fewer side effects than from other treatments [29]. abdominal massage is a professional massage that directly stimulates the digestive tract by increasing intra-abdominal pressures and bowel activity [30]. This induces rectal muscle waves that stimulate the somato-autonomic reflex and bowel sensation to promote rectal loading and peristalsis [31]. abdominal massage improves FC among the elderly and individuals with multiple sclerosis, spinal cord injuries and profound disability [12,28,32,33]. However, AM effects remain controversial due to variations in treatment durations, pressures and patient characteristics and receptivity.

Evidence suggests ES improves FC; however, its mechanisms of action is unknown. Transcutaneous Electrical Nerve Stimulation (TENS) and Interferential Current (IFC) therapy are theorized to send targeted electrical currents to spinal ganglion areas that regulate peristalsis to accelerate colonic transit and increase antegrade contraction frequency [5,34,35,36]. Using ES over bowel tissues may cause reflex inhibition of Sympathetic Nervous System (SNS) activity that inhibits bowel mobility and motility [35]. However, further study is required to determine ES mechanisms of action and optimal dosing parameters for various colonic diseases.

There are no current studies on the use of MLD for FC. The Vodder MLD technique was developed by a PT to facilitate lymphatic flow [37]. Currently, MLD is used to treat lymphedema following cancer treatments and to reduce pain and alleviate edema caused by venous diseases, surgery, primary lymphedema and trauma [5,37,38,39,40]. The MLD facilitates absorption of interstitial fluid into the lymph capillaries and lymphatic circulation. It also improves ANS balance by increasing Parasympathetic Nervous System (PNS) activity, which reduces physiologic stress and restores homeostasis [37]. Increased PNS activity promotes relaxation of intestinal sphincter muscles and accelerates peristalsis which can relieve FC [41,42,43]. An ANS imbalance is found with chronic Irritable Bowel Syndrome (IBS) which induces constipation or diarrhea [44,45]. The abdominal application of MLD is theorized to mitigate constipation by directly stimulating the bowel, presumably by reducing SNS activity, increasing PNS activity and restoring ANS balance. This treatment may relieve constipation and prevent recurrences since MLD provides direct stimulation to the bowel to promote motility and it may improve ANS balance [46,47,48]. Therefore, this study first investigated the effects of abdominal MLD on ANS balance, anxiety and stress and bowel movement frequency (BMF) and time–duration (BMT), then compared the outcomes to AM and ES. The study provides preliminary guidance to PTs on the use of MLD, AM and ES for managing FC and recommendations for future studies on these interventions.

## 2. Materials and Methods

### 2.1. Subjects

Following Institutional Review Board approval, investigators screened 366 college students using Rome III criteria to identify individuals with FC. Rome III criteria are used clinically to diagnose FC among individuals with IBS [49,50,51]. These criteria are valid and reliable with high sensitivity (ICC = 0.81), specificity (ICC > 0.99) and positive predictive value (ICC > 0.99) [52]. Individuals who met 2 of 6 Rome III symptoms of “loose stools are rarely present without the use of laxatives,” and “insufficient criteria for IBS for at least 12 weeks, not necessarily consecutive, in the past six months” were included in the study [51]. Exclusion criteria were IBS; OC or secondary constipation from neurogenic, metabolic, endocrine or postoperative diseases; cognitive impairment, aphasia or mental disorders; or severe liver, heart or kidney damage. Individuals were also excluded who: did not wish to participate; used cardiovascular drugs or anticoagulants: or implanted cardiac pacemakers [53]. Based on a screening questionnaire, 17 males and 38 females were initially selected, however 1 male withdrew for personal reasons resulting in 16 male and 38 female participants (Figure 1).

Participants received information about the study and completed informed consent. Participants were instructed to avoid any constipation medications such as enemas and laxatives during the study.

### 2.2. Interventions

Participants were age and gender matched, then randomly assigned to one of the three interventions: MLD, AM or ES. The frequency and duration of treatment was four-times a week for four weeks, like previous studies [5,9,33,35,53]. Treatments were performed in a quiet laboratory with the temperature maintained at 22 ℃–24 ℃ to minimize environmental effects on physiological responses. Participants were positioned in supine with a pillow under their knees to reduce lumbar extension and relax the abdomen. Movement and sleeping were avoided. Participants rested for 5 min and then received their assigned treatment for 15 min. Following treatments, participants rested for additional 5-min for a total intervention time of 25 min.

The principle investigators trained 3 assisting PTs in the treatment protocols to minimize experimental bias. The assisting PTs were blinded to the study’s purpose. The PT performing MLD was trained by a Dr. Vodder School certified MLD Instructor based on the Wittlinger et al. methods of using light pressure that compressed only the dermis [37]. This MLD protocol begins over the cervical lymph nodes and occiput, then continues over the abdominal area. During the abdominal MLD, subjects performed deep breathing. The 2 PTs performing AM and ES received training from PTs specializing in these professional techniques. The AM used moderate pressure that compressed the abdomen to stimulate the bowel in the direction of colonic movement, like methods used in prior studies [30,54,55].

The ES performed was IFC (KM-2900, Ulsan, Korea) using four vacuum electrodes with two channels of alternating currents applied in the side-lying position. Two pads were placed on the T9–L2 para-spinal area and two pads were placed diagonally on the anterior abdomen below the costal margin. A fixed frequency (4000 Hz) was used on one side, while the beat frequency was varied from 4000 to 4100 Hz by altering the frequency from 0 to 100 Hz on the alternate side. The intensity was <40 mA for subject comfort and to avoid muscle contractions, as in prior studies [3,4,34,35,56].

All participants were instructed to breathe naturally during Heart Rate Variability (HRV) measurement to avoid intrathoracic pressure changes that may affect the ANS study outcomes.

### 2.3. Measurement

Measures were taken prior to and one week following the four weeks of treatment. Heart Rate Variability measures assessed ANS activity. Electrodes, attached to the right and left wrists, recorded ECG signals using WEEG32 (LAXTHA, Daejeon, Korea). The ECG signals were converted from 256 Hz to digital signals using a 12-bit analog-digital converter and then stored on a laptop computer for frequency analyses with the TeleScan software package (LAXTHA). Digitized ECG signals identified R-wave peaks for beat-to-beat heart rate interval calculations (i.e., R-R intervals). Spectral analysis measured the R-R intervals and generated HRV measures in 3 main frequency bands: total power (TP), for overall ANS activity; low frequency (LF) power, for SNS activity sampled from 0.04–0.15 Hz; and high frequency (HF) power, for PNS activity sampled from 0.15–0.40 Hz. The LF and HF were converted to normalized numbers and reported as LFnu and HFnu. The TP may be expressed as absolute values or normalized units, whereas LF and HF are generally expressed as normalized units to assess ANS activity [37]. For clinical relevance, LFnu needs to decrease by at least 0.66 Hz and HFnu needs to increase at least 0.44 Hz. to signify increases in PNS activity and improvements in ANS balance [57]. Normalized units were reported for spectral indices and calculated as: LFnu = LF/(LF + HF); and HFnu = HF/(LF + HF) [58]. Finally, the LF/HF ratio was calculated with higher PNS activity and lower SNS activity indicated by increases in this ratio which indicates improved ANS balance and reduced physiologic stress [59,60,61].

The state-trait anxiety inventory (STAI), developed by Spielberger [62] and translated then modified by Kim Jeong Taek into Korean, measured participants’ anxiety [60,61,62,63]. This self-reported 20-item questionnaire asks how an individual feels at a specific moment in a situation. Items are rated on a four-point scale and then summed to compute the total score, ranging from 20–80 (Table 1).

A prior study found a mean score of 42.5 among 816 college students which was within the normal range which indicates STAI is recommended for use among college students [63]. The stress response inventory (SRI) measured the subjects’ degree of stress. The SRI is a 39-item scale developed by Koh et al. for use among normal people [64]. Each item is rated on a five-point scale. Internal consistency, concurrent validity and test-retest reliability were ICC = 0.97, ICC = 0.54–0.76 and ICC = 0.69−0.96, respectively. Sensitivity, specificity and positive predictive values were ICC = 0.57, ICC = 0.74 and ICC = 0.71, respectively or individuals 20 years of age or older [64].

Participants kept daily journals of their bowel movement frequency (BMF) per day and time duration (BMT) in minutes to monitor their progress beginning one week before and continuing through one week after the interventions.

### 2.4. Statistics

Data are expressed as the mean (SD) values. The one-sample Kolmogorov–Smirnov test assessed normality for all variables. The paired samples *t*-test assessed differences between pre-and posttest measures within groups. One-way ANOVA with post hoc Sheffe’s analysis compared differences between group outcomes. Secondary analysis with Pearson correlation coefficients examined associations between ANS measures, psychological factors and bowel movement journal entries. The statistical packages for the social sciences (SPSS ver. 21.0) analyzed the data using two-tailed probability with p < 0.05 considered statistically significant.

## 3. Results

Subject characteristics are reported in Table 2 and there were no significant differences between groups pretreatment.

### 3.1. Autonomic Nervous System Measures

Table 3 shows the ANS results with TP significantly increased from MLD (p < 0.01), but not AM or ES. Following MLD, TP increased 671.97 Hz. which exceeded the minimal detectable difference (MDD) calculated as 2.44 Hz. and represents a positive shift toward improved ANS balance.

The minimal clinically important difference (MCID) for this HRV measure is not reported in the literature. The LFnu and LF/HF ratios significantly decreased and HFnu significantly increased from MLD (p < 0.01) and AM (p < 0.04), but not ES. The LFnu, HFnu and LF/HF ratio improvements exceeded MDDs of 1.71, 1.51 and 0.15, respectively following MLD and AM (Table 4). Further, the LF reductions of 4.32 Hz. and 3.11 Hz. and HF increases of 4.33 Hz. and 3.11 Hz. for MLD and AM, respectively, exceeded the MCID for an LF decrease of 44 Hz and an HF increase of 66 Hz. Improvements in the LF/HF ratio exceeded the calculated MCID of 66. These changes signified improvement in ANS balance through increases in PNS activity. Table 3 shows between group differences in ANS measures which found TP was significantly higher following MLD compared to both AM and ES (p < 0.04). However, no significant differences were found between groups for LFnu, HFnu and LF/HF ratios (Table 4).

The calculated effect sizes for between group measures were small, at 251 and 291 with power at 34 and 48, respectively. Therefore, a future investigation would require a sample of 40–50 subjects in each group for adequate statistical power of these ANS measures.

### 3.2. Changes in Anxiety and Stress Measures

Reductions in STAI anxiety measures within each group were highest following MLD; however, these improvements did not reach statistical significance (p = 0.6). The differences between groups were also not significant (p = 0.74) (Table 5).

Reductions in stress measures within all groups reached statistical significance following MLD (p < 0.1), AM (p < 0.4) and ES (p < 0.1). The largest reduction was 10.05 following MLD compared with 3.45 for AM and 2.0 for ES. The MLD improvements exceeded the MDD of 8.62; however, this reduction from 90.55 to 81.50 remained in the “High stress, need to manage stress” category score of 81–120. Additionally, differences between groups for SRI measures were not significant (p = 0.6) (Table 5). The calculated effect sizes for between group anxiety and stress measures were small at 33 with power at 54. Therefore, a future investigation would require a sample of 40 subjects in each group for adequate statistical power of these measures.

### 3.3. Changes in Bowel Movement Frequency and Time Duration

The BMF increased significantly following MLD (p < 0.01) and AM (p < 0.01), but not after ES (p = 0.08). The BMT also decreased significantly following MLD (p = 0.01), AM (p = 0.02) and ES (p < 0.01). The MLD had the highest reduction of 2.72 min compared with AM of 1.83 min and ES of 2.0 min. However, no significant differences were found between groups for either the BMF (p = 0.34) or the BMT (p = 0.34) (Table 6).

### 3.4. Correlations between ANS, BMF and BMT, Stress and Anxiety

Table 7 shows the HFnu increases which indicate improved PNS activity, had significant negative correlations with reductions in BMT (r = 0.96), stress (r = 0.64) and anxiety (r = 0.52) and a significant positive correlation with increased BMF (r = 0.59).

The LF/HF ratio reductions which represent improved ANS balance, had significant positive correlations with BMT (r = −0.61), anxiety (r = −0.53) and stress (r = −0.60) and a significant negative correlation with BMF (r = −0.58). There is a significant positive correlation between BMT and stress (r = −0.55) and a significant negative correlation between BMF and stress (r = −0.47).

## 4. Discussion

This single-blind randomized controlled study examined the effects of MLD on FC by measuring changes in ANS balance, anxiety and stress and bowel movement frequency and time–duration. The MLD outcomes were then compared to those of AM and ES. The effects of AM and ES on FC were previously studied independently, which did not allow direct comparisons of their outcomes. No prior studies were found that scientifically examined the effects of MLD on FC. However, it was theorized that MLD would improve FC since it could be performed on the abdomen and because it may normalize ANS balance to promote improved gastrointestinal function. This study was performed on college-aged adults in their 20s with FC symptoms.

Participants were age and gender matched, then randomly assigned to one of the three interventions. The study design used pre-and post-intervention measures which allowed direct comparisons of outcomes between the three interventions.

The ANS mediates bidirectional brain–gut interactions [31,47]. Therefore, methods that improve ANS balance may restore or normalize intestinal function [31,47]. While no studies were found that reported ANS measures for college-aged students with FC, there is evidence that individuals with IBS-C exhibit an ANS imbalance from increased SNS activity and decreased PNS function [65,66]. The ANS balance in this study was assessed using HRV measures of TP, HFnu, LFnu and LF/HF ratios, which are valid and reliable measures [67,68]. In only the MLD group, TP improved significantly and clinically representing improved ANS balance. The LFnu, HFnu and LF/HF ratios improved significantly and clinically in the MLD and AM groups, while only LFnu improved in the ES group. Since people with IBS have reductions in the ANS balance, these improvements can be interpreted overall as a positive outcome [67]. Overall, the MLD group had significantly higher improvements in TP than the AMA and ES groups, and larger, but not statistically significant improvements, for HFnu, LFnu and LF/HF ratios compared with AM or ES. These results are the first to imply MLD better improved PNS activity and overall ANS balance compared to the other treatments.

The improved outcomes for MLD compared to AM on ANS measures may be due to differences between the protocol methods. First, MLD begins at the neck, in the area over the vagus nerve, then it proceeds to the descending colon and finally, is performed over the ascending and transverse colon [37]. Thus, beginning treatment over the vagus nerve may initiate or enhance ANS activity prior to beginning MLD over the abdomen. Second, the gentle pressures used to the dermis over the abdomen during MLD may not cause abdominal or thoracic cavity pressure changes like the moderate pressures used in AM. However, the subjects actively engage in deep breathing during abdominal MLD which may internally stimulate abdominal and thoracic cavity pressure changes to promote colonic motility and peristalsis. Thus, the combination of vagus nerve stimulation and the subject’s deep breathing during MLD abdominal procedures, may provide the superior changes in ANS balance.

Gastrointestinal (GI) functions are influenced by PNS activity, and stimulation of the PNS through MLD, AM and ES are believed to provide FC relief by increasing intestinal motility and digestive secretions and relaxing GI canal sphincters [69]. Since MLD and AM are direct-types of therapy designed specifically to target the digestive tract, this may explain why they had better effects than ES on most study measures [30]. Alternatively, ES provides indirect stimulation and further study is required to determine its mechanisms of action and optimal dosing for frequency, waveform, site, intensity and treatment durations in various populations.

Although, not previously studied among college-aged adults with FC, anxiety and mental stress contribute to pathogenic factors that trigger and maintain GI symptoms in people with IBS [70,71,72]. The STAI anxiety measures improved slightly following each intervention, but changes were not significant either within or between groups. While reductions in anxiety were highest from MLD compared to AM or ES, all pre-and post-measures remained in the “mildly high state” anxiety category. The lack of statistical significance between groups may require more sensitive measurement tools, longer treatments durations and follow-up times or a larger sample. Alternatively, changes in SRI stress measures were statistically higher within all groups, but not significantly different between groups. These results suggest that all interventions produced sufficient comfort and relaxation to reduce psychological stress among people with FC [13]. These interventions may increase the body’s vitality by improving homeostasis and elevating PNS activity. Since the PNS contains approximately 80% of sensory fibers, this increase in feelings of well-being may indirectly contribute to positive psychological changes.

For the bowel movement journal entries, MLD and AM groups had statistically significant increases in BMF, and all three groups had significant increases in BMT. However, between group differences were again not significant possibly due to the small sample. Although not previously studied, MLD was found in this study to significantly improved BMF and BMT which is associated with reduced constipation. Although, effects of AM and ES on BMF and BMT were extensively studied, the results remain controversial as some studies found positive outcomes while others did not [2,12,32,33,73]. Hence, more controlled trials are required to determine the effects of AM and ES for various colon diseases in different populations. For adults in their 20s with FC, this study found that AM was effective on both BMF and BMD, while ES was effective on BMF alone.

Abdominal massage effects on FC are controversial. Kim et al. reported constipation was reduced after 10 days among older people, whereas another study reported it was not effective even after 4 weeks of treatment [54,74]. However, in the current study, a four-day-a-week, four-week intervention was found effective for constipation relief. Both MLD and AM are considered direct stimulation methods, although MLD uses light pressure to the dermis over the abdomen, while AM uses moderate pressure to the abdomen. Both techniques are thought to move stool manually along the digestive tract and to increase intestinal motility by stimulating the somato-autonomic reflex that mitigates tension, thereby alleviating constipation [75]. Previous studies also found that AM with both light and moderate pressures was effective [13,28,33,54,74].

The study initially examined the primary measures independently, but ANS activity and psychological factors appear to have related and coordinated effects on bowel movement factors. First, the ANS links the central nervous system and gut, and ANS dysfunction alters normal bowel habits [45,61]. In addition, anxiety and mental stress contribute to elevated cortisol concentrations which advance insulin resistance to hyperinsulinemia, and insulin promotes SNS activity [76,77]. Evidence finds that IBS-C is associated with elevated insulin levels and activation of SNS activity which affects bowel movement characteristics [78]. Additionally, chronic or excessive stress induces chronic vagal suppression which increases SNS activity [70]. Therefore, the study also examined correlations between overall ANS improvements with anxiety, stress and with BMF and BMT following the interventions.

Increases in HFnu, signifying increased PNS activity, significantly correlated with reductions in stress, anxiety and BMT and with increased BMF. Reductions in the LF/HF ratio, signifying improved ANS Balance, significantly correlated with reductions in BMT, stress and anxiety and increased BMF. The shorter BMT significantly correlated with lower stress and anxiety and increased BMF. Like previous studies, the present study supports correlations between increased PNS activity and lower constipation and anxiety [79,80]. The elevated HFnu, signifying lower SNS activity and the reduced LF/HF ratio following MLD, AM—and to a lesser extent ES—support an increase in PNS activity and restoration of ANS balance. These results support ANS improvements from the study interventions were ultimately correlated with improvements in stress and anxiety and mitigation of constipation symptoms.

This study limitations include: the small sample size which may have contributed to the lack of statistical significance for between groups measures; There was a known wide variability and ambiguity in each individual’s definition of constipation and loose stools; FC occurs more frequently among women, however both men and women were included in this study, so a future study of men and women separately would provide better generalizability; and it was not possible to determine any short or long-term outcomes from the treatments since the final measures were taken one week after the interventions. Therefore, based on study findings, future studies may wish to examine differences between genders, ages, longer treatment durations, short and long-term treatment effects and individuals with various colon diseases. Future studies could also examine the effects of combining these interventions to determine if there are earlier effects or longer lasting improvements.

## 5. Conclusions

The study found MLD, AM and ES, to a lesser extent, increased PNS activity and lowered the sympathetic/parasympathetic ratio, which improved ANS balance and alleviated constipation in adults in their 20s with FC. The MLD had more positive effects on improvements of ANS, constipation relief and anxiety and stress compared with AM and ES. Furthermore, these interventions had positive effects on participant’s anxiety and stress measures.

## Figures and Tables

**Figure 1 ijerph-17-03924-f001:**
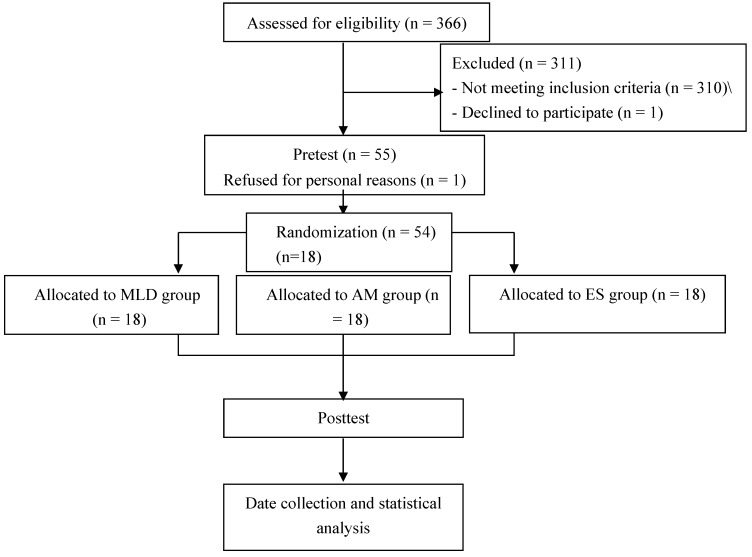
Experimental procedures time flow chart. MLD: manual lymph drainage, AM: abdominal massage, ES: electrical stimulation.

**Table 1 ijerph-17-03924-t001:** Psychological Inventories Descriptors.

State-Trait Anxiety Inventory (STAI) Category		Stress Response Inventory (SRI) Category	
Score	Description	Score	Description
<53	Within normal range	≤50	Low stress
52–56	Mildly high state anxiety	51–80	Need mind control
57–61	Fairly high state anxiety	81–120	High stress, need to manage stress
≥62	Severely high state anxiety	≥120	Severe stress, need counseling

**Table 2 ijerph-17-03924-t002:** General characteristics of subjects (n = 54).

	MLD (n = 18)	AM (n = 18)	ES (n = 18)	p ≤ 0.5
Sex				
Male/Female	5/13	5/13	6/12	NS
Age (year)	21.61 ± 2.35	23.72 ± 2.02	23.38 ± 1.28	NS
Height (cm)	166.24 ± 7.67	164.42 ± 6.56	164.74 ± 7.88	NS
Weight (kg)	63.17 ± 12.35	61.92 ± 7.76	62.05 ± 8.59	NS
BMI (kg/m^2^)	22.72 ± 2.97	22.87 ± 2.20	22.76 ± 1.64	NS

MLD: manual lymph drainage, AM: abdominal massage, ES: electrical stimulation, BMI: body mass index. NS: not significant.

**Table 3 ijerph-17-03924-t003:** Autonomic nervous system *t*-test (n = 54).

		TP	LF	HF	LF/HF Ratio
MLD	Pre	1306.43 (613.22)	64.38 (6.11)	35.61 (6.11)	1.89 (0.51)
	post	1978.40 (710.74)	60.06 (6.67)	39.94 (6.67)	1.56 (0.42)
	t	−4.51	2.75	−2.75	2.37
	p	0.01 *	0.01 *	0.01 *	0.01 *
AM	Pre	1426.93 (932.40)	65.00 (6.15)	35.00 (6.15)	1.94 (0.57)
	post	1611.51 (818.19)	61.89 (4.87)	38.11 (4.87)	1.66 (0.34)
	t	−1.56	2.13	−2.13	2.22
	p	0.14	0.04 *	0.04 *	0.04 *
ES	Pre	1357.58 (1062.42)	64.44 (5.47)	35.56 (5.47)	1.88 (0.45)
	post	1361.23 (670.15)	62.72 (4.81)	37.28 (4.81)	1.73 (0.36)
	t	−0.02	−2.75	−1.02	1.12
	p	0.98	0.01 *	0.32	0.28

TP: total power, LFnu: low frequency (normalized units), HFnu: high frequency (normalized units), MLD: manual lymph drainage, AM: abdominal massage, LF/HF ratio: ratio of low frequency (LF) to high frequency (HF), ES: electrical stimulation. *: significant difference (p < 0.05).

**Table 4 ijerph-17-03924-t004:** ANOVA analyses of autonomic nervous system measures (n = 54).

	MLD (n = 18)	AM (n = 18)	ES (n = 18)	*F*	p
TP	1978.40 (710.74)	1611.51 (818.19)	1361.23 (670.15)	3.21	0.4 *
LFnu	60.06 (6.67)	61.89 (4.87)	62.72 (4.81)	1.10	3.4
HFnu	39.94 (6.67)	38.11 (4.87)	37.28 (4.81)	1.09	3.4
LF/HF ratio	1.56 (0.42)	1.66 (0.34)	1.73 (0.36)	0.82	4.5

TP: total power, LFnu: low frequency (normalized units), HFnu: high frequency (normalized units), MLD: manual lymph drainage, AM: abdominal massage, LF/HF ratio: ratio of low frequency (LF) to high frequency (HF), ES: electrical stimulation. *: significant difference (p < 0.05) between MLD group and both AM and ES group.

**Table 5 ijerph-17-03924-t005:** Anxiety and stress *t-test* and ANOVA analyses (N = 54).

	MLD (n = 18)		AM (n = 18)		ES (n = 18)		*F*	p
	Pre	Post	Pre	Post	Pre	Post		
STAI	56.00 ± 7.67	53.61 ± 6.56	55.61 ± 8.07	55.33 ± 7.81	55.17 ± 7.59	54.56 ± 5.20	0.31	0.74
*t*	1.99		0.28		0.53			
*p*	0.06		0.78		0.60			
SRI	90.55 ± 15.99	81.50 ± 14.19	89.00 ±1 8.33	85.55 ± 21.04	90.72 ± 15.02	86.22 ± 13.91	0.42	0.66
*t*	3.09		2.29		3.11			
*p*	0.01 *		0.04 *		0.01 *			

MLD: manual lymph drainage, AM: abdominal massage, ES: electrical stimulation. SRI: Stress response inventory, STAI: state-trait anxiety inventory. *: significant difference (p < 0.05) between before and after experiment.

**Table 6 ijerph-17-03924-t006:** Bowel movement frequency and time–duration *t-test* and ANOVA analyses (N = 54).

	MLD (n = 18)		AM (n = 18)		ES (n = 18)		*F*	p
	Pre	Post	Pre	Post	Pre	Post		
Frequency/a week	3.11 (1.28)	4.17 (1.25)	2.84 (1.10)	3.83 (1.42)	3.01 (0.94)	3.56 (1.29)	0.96	0.39
t	−4.78		−4.37		−1.84			
*p*	0.01 *		0.01 *		0.08			
Time Duration **	10.11 (4.44)	7.39 (3.97)	10.72 (4.44)	8.89 (4.45)	11.11 (4.75)	9.11 (3.60)	0.96	0.39
t	3.84		2.50		3.77			
*p*	0.01 *		0.02 *		0.01 *			

MLD: manual lymph drainage, AM: abdominal massage, ES: electrical stimulation. *: significant difference (p < 0.05) between before and after experiment. **: minutes.

**Table 7 ijerph-17-03924-t007:** Pearson correlation coefficients between autonomic balance, bowel movement frequency and time–duration and psychological factors. (N = 54).

	HF (Post-pre)	LF/HF Ratio (Post-pre)	Time Duration (Post-pre)	Frequency (Post-pre)	SRI (Post-pre)	STAI (Post-pre)
HF (Post-pre)	1	−0.96 *	−0.66 *	0.59 *	−0.64 *	−0.52 *
LF/HF Ratio (Post-pre)		1	0.61 *	−0.58 *	0.60 *	0.53 *
Time Duration (Post-pre)			1	−0.44 *	0.55 *	0.23
Frequency (Post-pre)				1	−0.47 *	−0.29 *
SRI (Post-pre)					1	0.23
STAI (Post-pre)						1

* Statistically significant at the level of p < 0.5. HFnu: high frequency in normalization. LFnu: low frequency in normalization. SRI: Stress response inventory. STAI: state trait–anxiety inventory.

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
