# Peer review of "Comparisons between Manual Lymph Drainage, Abdominal Massage, and Electrical Stimulation on Functional Constipation Outcomes: A Randomized, Controlled Trial"

_ijerph, 2020, doi:10.3390/ijerph17113924_

Round 1

Reviewer 1 Report

Thank you for the opportunity to review this manuscript. I have a few concerns.

  1. Most of the citations in the background describing support for AM and ES to treat constipation refer to a pediatric population. However, the sample is adult. Is there data supporting your comments in adult population. If no, this may be used as the gap in the literature.
  2. The background does not mention dyssynergic defecation. Excluding mechanical obstruction reason for constipation, DD is the primary reason for constipation. DD has a higher prevalence in females and younger age individuals. I suggest adding this to your background. https://www.ncbi.nlm.nih.gov/pubmed/18793997
  3. Neuro stimulation for neuropathy relative fecal incontinence literature exists. I suggest mention this data in your background to strengthen ES for pelvic floor issues. https://www.ncbi.nlm.nih.gov/pmc/articles/PMC4019944/
  4. Exclusion criteria: it does not mention opioid prescription? Does the sample include subjects with a prn or chronic opioid prescription. OP prescription impacts the likelihood for chronic constipation. https://journals.lww.com/ajg/Fulltext/2019/11000/Is_Dyssynergic_Defecation_an_Unrecognized_Cause_of.16.aspx
  5. Curious why PT were trained to perform AM instead of hiring certified massage therapist? I suggest insert a citation indicating a PT and MT deliver standard care.
  6. For bowel variables, why did you not use the Bristol Stool Scale. This scale is the standard method of evaluating stool formation to qualitatively assess bowel disorders.
  7. Since the group is young, less than 25 yoa, why are their SRI scores so high? Did the study take place during finals?
  8. I am not sure if these folks are FC? Their bowel frequency is quite high, 2.8 - 4.2/wk. The Bristol Stool Scale would justify these metrics.
  9. How did deep breathing techniques in AM and MLD impact the results, confounding variable?

Author Response

Thank you very much for your detailed review and comments. The answer to your review is as follows.

1. Most of the citations in the background describing support for AM and ES to treat constipation refer to a pediatric population. However, the sample is adult. Is there data supporting your comments in adult population. If no, this may be used as the gap in the literature.

Reply: Researchers know that most of the previous AM studies cited in this study were conducted mainly for adults and the elderly, and ES was studied for infants by Yik YI and Clarke MCC, but studies on adults are still insufficient. I agree that there may be a gap with previous studies of children in interpreting results for ES. However, in the future, it is believed that this study, in which other groups of ES researchers conducted ES on adults, will be used as valuable data.

2. The background does not mention dyssynergic defecation. Excluding mechanical obstruction reason for constipation, DD is the primary reason for constipation. DD has a higher prevalence in females and younger age individuals. I suggest adding this to your background. https://www.ncbi.nlm.nih.gov/pubmed/18793997

Reply:  Thank you for the new good information. I have inserted in the background the contents of the dyssynergic defecation you have presented with a quotation.

3. Neuro stimulation for neuropathy relative fecal incontinence literature exists. I suggest mention this data in your background to strengthen ES for pelvic floor issues. https://www.ncbi.nlm.nih.gov/pmc/articles/PMC4019944/

Reply: I have inserted the information you gave me into the background. Thank you.

4. Exclusion criteria: it does not mention opioid prescription? Does the sample include subjects with a prn or chronic opioid prescription. OP prescription impacts the likelihood for chronic constipation. https://journals.lww.com/ajg/Fulltext/2019/11000/Is_Dyssynergic_Defecation_an_Unrecognized_Cause_of.16.aspx

Reply:  Opioid prescription is very rare in Korea.  The exclusion criteria were therefore excluded. Thank you for a good comment.

5. Curious why PT were trained to perform AM instead of hiring certified massage therapist? I suggest insert a citation indicating a PT and MT deliver standard care.

Reply:  Unfortunately, Korea does not have a certified massage therapist system. Only Physical therapist can legally perform a massage medically.

6. For bowel variables, why did you not use the Bristol Stool Scale. This scale is the standard method of evaluating stool formation to qualitatively assess bowel disorders.

Reply: Thank you for your good comment. I totally agree with you. There were some difficulties in the progress of the experiment to evaluate the Bristol Stool Scale in our study. I will reflect it in the next study.

7. Since the group is young, less than 25 yoa, why are their SRI scores so high? Did the study take place during finals?

Reply: It is believed that the test period was affected. It is very surprising to see the details of the judges. Thank you.

8. I am not sure if these folks are FC? Their bowel frequency is quite high, 2.8 - 4.2/wk. The Bristol Stool Scale would justify these metrics.

Reply: I agree very much that the bowel frequency is rather high. We will consider using The Bristol Stool Scale for further research.

9. How did deep breathing techniques in AM and MLD impact the results, confounding variable?

Reply:  Deep breathing was not implemented when AM was implemented. However, he mentioned it in the discussion, thinking that deep breathing would affect the results of the application of MLD to promote the flow of abdominal lymph nodes.

Thank you for your very delicate review. We are judged to be a better paper by your comments.

Reviewer 2 Report

There are numerous problems with this manuscript. The authors need to do a close edit and a serious revision of the manuscript. 

Specific comments:

  1. I do not think IJERPH is the right journal for this paper; it should be submitted to a complementary and alternative medicine-themed journal instead, or Gastrointestinal Disorders.
  2. It is a "randomized, controlled trial", not "Random Controlled Trial".
  3. Please report the exact P-values rather than simply P<0.02 or P<0.01. Statistical results should include confidence intervals or exact P values, even for non-significant results. All statistical results should be reported according to the SAMPL (Statistical Analyses and Methods in the Published Literature) Guidelines.
  4. The definition for FC given in the introduction is grossly incomplete. Duration of symptoms and not meeting criteria for IBS (no loose stools) are important to mention as well. Please cite the appropriate Rome IV criteria for FC.
  5. The American Gastroenterological Association (AGA) and Rome III criteria both emphasize the need to identify defecatory disorders. This is not considered in the current study. In fact, in the last AGA technical review, the term “functional constipation” has been dropped because a subset of patients with symptom criteria for functional constipation have slow colonic transit. Moreover, in several small studies, slow transit constipation (STC) was associated with a marked reduction in colonic intrinsic nerves and interstitial cells of Cajal, that is, it is not truly a functional disorder (citation: ncbi.nlm.nih.gov/pmc/articles/PMC3531555).
  6. Why did the study investigators adopt Rome III and not Rome IV criteria, which were published in 2016?
  7. How was the sample size determined? There is currently no evidence of power calculation. The present sample appears small and limited to a convenience sample.
  8. CONSORT stands for Consolidated Standards of Reporting Trials and encompasses various initiatives developed by the CONSORT Group to alleviate the problems arising from inadequate reporting of randomized controlled trials. It is unclear if these guidelines were followed. Please specify. The authors should report this trial in accordance with CONSORT guidelines.
  9. Was the study protocol prospectively registered?
  10. Please change "table 2" to "Table 2".
  11. Please change "Table 4. Autonomic nervous system ANOVA analyses" to "Table 4. ANOVA analyses of autonomic nervous system measures".
  12. Be consistent with numbers of >10, when they are not starting a sentence; they can be written in full, or written as a number, but not both. Be consistent.
  13. When the authors write that "Journals were reviewed by research investigators for accuracy and completeness", what exactly do you mean by accuracy? There is a known wide variability and ambiguity in each individual’s definition of constipation and loose stools (citation: ncbi.nlm.nih.gov/pubmed/28960557).
  14. Please change "single-blind random controlled study" to "single-blind, randomized controlled trial".
  15. In addition to ref 70 and 71, suggest authors cite a very recent meta-analysis supporting the association between PTSD and IBS symptoms (citation: ncbi.nlm.nih.gov/pubmed/30144372).
  16. The statement "a future study that examined men and women separately would provide better generalizability" is erroneous. 

Author Response

Thank you very much for your detailed review and comments. The answer to your review is as follows.

  1. I do not think IJERPH is the right journal for this paper; it should be submitted to a complementary and alternative medicine-themed journal instead, or Gastrointestinal Disorders.

Reply:  Thank you for recommending the journal.

  1. It is a "randomized, controlled trial", not "Random Controlled Trial".

Reply: I revised it according to your opinion

  1. Please report the exact P-values rather than simply P<0.02 or P<0.01. Statistical results should include confidence intervals or exact P values, even for non-significant results. All statistical results should be reported according to the SAMPL (Statistical Analyses and Methods in the Published Literature) Guidelines.

Reply: I totally agree with you. The correct P-value was presented.

  1. The definition for FC given in the introduction is grossly incomplete. Duration of symptoms and not meeting criteria for IBS (no loose stools) are important to mention as well. Please cite the appropriate Rome IV criteria for FC.

Reply:  The definition of FC generally describes the methods defined in the study. Thank you for your comment. I understand that research using Rome III has been conducted until recently. In future research, we will proceed with research using Rome IV criteria. Thank you for the good information.

  1. The American Gastroenterological Association (AGA) and Rome III criteria both emphasize the need to identify defecatory disorders. This is not considered in the current study. In fact, in the last AGA technical review, the term “functional constipation” has been dropped because a subset of patients with symptom criteria for functional constipation have slow colonic transit. Moreover, in several small studies, slow transit constipation (STC) was associated with a marked reduction in colonic intrinsic nerves and interstitial cells of Cajal, that is, it is not truly a functional disorder (citation: ncbi.nlm.nih.gov/pmc/articles/PMC3531555).

Reply:   Your comments are wonderful and thank you very much. In the next study, we will consider the knowledge and contents provided.

  1. Why did the study investigators adopt Rome III and not Rome IV criteria, which were published in 2016?

 Reply:  We conducted a study using Rome III in the previous preliminary study and based on this study. We will make sure that the research reflecting the Rome IV can be carried out next time. Thank you very much for the good comment.

  1. How was the sample size determined? There is currently no evidence of power calculation. The present sample appears small and limited to a convenience sample.

Reply:   The sample size could not be determined according to power calculation. I regret the small number of samples as you suggested. Please understand that it is a matter of our research period, financial problem, and research environment.

  1. CONSORT stands for Consolidated Standards of Reporting Trials and encompasses various initiatives developed by the CONSORT Group to alleviate the problems arising from inadequate reporting of randomized controlled trials. It is unclear if these guidelines were followed. Please specify. The authors should report this trial in accordance with CONSORT guidelines.

Reply:   We proceeded according to CONSORT guideline. Thank you.

  1. Was the study protocol prospectively registered?

Reply:  This research was conducted with IRB approval and pre-reported to the Clinical Research Information Service (CRIS).

  1. Please change "table 2" to "Table 2".

Reply:   Changed it. Thank you.

  1. Please change "Table 4. Autonomic nervous system ANOVA analyses" to "Table 4. ANOVA analyses of autonomic nervous system measures".

Reply:  Changed it. Thank you.

  1. Be consistent with numbers of >10, when they are not starting a sentence; they can be written in full, or written as a number, but not both. Be consistent.

Reply:  Checked it and revised it.

  1. When the authors write that "Journals were reviewed by research investigators for accuracy and completeness", what exactly do you mean by accuracy? There is a known wide variability and ambiguity in each individual’s definition of constipation and loose stools (citation: ncbi.nlm.nih.gov/pubmed/28960557).

Reply:  Accuracy means that researchers have organized their own bowel movements for statistical analysis. We agreed with you and deleted the words from the sentence.

  1. Please change "single-blind random controlled study" to "single-blind, randomized controlled trial".

Reply:  changed it. Thank you

  1. In addition to ref 70 and 71, suggest authors cite a very recent meta-analysis supporting the association between PTSD and IBS symptoms (citation: ncbi.nlm.nih.gov/pubmed/30144372).

Reply:  added it. Thank you

  1. The statement "a future study that examined men and women separately would provide better generalizability" is erroneous. 

 Reply:  There was some ambiguity in the sentence, so the sentence was modified. Thank you

Round 2

Reviewer 2 Report

  1. Minimal changes were made by the authors. It is a "randomized, controlled trial", not "randomed, controlled trial".
  2. The investigators, unfortunately, chose to reply nicely but did not address all of the important concerns. There is a known wide variability and ambiguity in each individual’s definition of constipation and loose stools (citation: ncbi.nlm.nih.gov/pubmed/28960557), which should be discussed as a study limitation.
  3. The placebo response is potentially greatest for the chronic, relapsing, and subjective pain of an illness such as functional constipation (citation: ncbi.nlm.nih.gov/pmc/articles/PMC4146777). Not only do the symptoms wax and wane, but diagnosis, reassurance, and empathy enhance the placebo effect. There was no provision for this in this study.

Author Response

Response to the reviewer’s comments

First of all, I would like to thank you sincerely for your detailed comments. We tried to revise our manuscript according to possible reviewers' comments. Even if there is something lacking, please understand. The answer to the second comment is as follows.

  1. Minimal changes were made by the authors. It is a "randomized, controlled trial", not "randomed, controlled trial".

Answer: Thank you for your checking it. We changed it.

  1. The investigators, unfortunately, chose to reply nicely but did not address all of the important concerns. There is a known wide variability and ambiguity in each individual’s definition of constipation and loose stools (citation: ncbi.nlm.nih.gov/pubmed/28960557), which should be discussed as a study limitation.

Answer: Thank you for your kind comment. We tried to revise the points you had commented. We discussed about your comment as a study limitation.

  1. The placebo response is potentially greatest for the chronic, relapsing, and subjective pain of an illness such as functional constipation (citation: ncbi.nlm.nih.gov/pmc/articles/PMC4146777). Not only do the symptoms wax and wane, but diagnosis, reassurance, and empathy enhance the placebo effect. There was no provision for this in this study.

Answer: We totally agree with your opinion. We discussed the need for the No intervention group to find out about the placebo effect. However, we didn't create a placebo group because we thought it was better to secure more samples of the experimental groups. I think this is also a limitation of research. In the next study, we will also design experiments to obtain results for placebo effects.